# Application Value of Systemic Inflammatory Indexes in the Clinical Evaluation of Patients with Heart Failure with Preserved Ejection Fraction (HFpEF)

**DOI:** 10.3390/medicina58101473

**Published:** 2022-10-17

**Authors:** Ruxin Wang, Juan Wu, Haowen Ye, Xiaofang Zhang, Lihong Wang

**Affiliations:** 1Department of Endocrinology and Metabolism, The First Affiliated Hospital of Jinan University, Guangzhou 510630, China; 2Clinical Laboratory, Suqian First People’s Hospital Affiliated to Nanjing Medical University, Suqian 223812, China; 3Clinical Experimental Center, The First Affiliated Hospital of Jinan University, Guangzhou 510630, China

**Keywords:** HFpEF, systemic inflammatory indexes (SII), lymphocyte-to-monocyte ratio (LMR), NT-proBNP, New York Heart Association (NYHA)

## Abstract

*Background:* In areas where medical resources are scarce, an economical and convenient way to assess patients’ condition so that treatment plans can be adjusted in a timely manner makes sense. The clinical value of systemic inflammatory indexes (SII) such as neutrophil-to-lymphocyte ratio (NLR), lymphocyte-to-monocyte ratio (LMR), albumin-to-gamma-glutamyl-transferase ratio (AGR), white-blood-cell-count-to-mean-platelet-volume ratio (WMR), high-density-lipoprotein-cholesterol-to-C-reactive-protein ratio (HCR), etc. were explored in heart failure (HF) with preserved ejection fraction (HFpEF) because of their easy availability and clinical value in the diagnosis, therapy and prognosis of cardiovascular diseases. *Methods:* 189 inpatients (including 48 patients with New York Heart Association (NYHA) I in the control group, and 141 patients with NYHA II-IV in the study group) from The First Affiliated Hospital of Jinan University, during the period July 2018 to March 2022, were included by retrieving electronic medical records. Logistic regression analysis, Spearman’s correlation coefficient, operating characteristic curve, etc. were used to analyze the data. *Results:* In patients with HFpEF, LMR (OR = 0.463, 95% CI 0.348–0.617, *p* = 0.000), NLR and N-terminal pro-B-type natriuretic peptide (NT-proBNP) were independent predictors for the presence of HF, and LMR (OR = 2.630, 95% CI 2.016–3.435, *p* = 0.000), NLR, FAG, MHR, AGR and NT-proBNP were independent predictors for increased NYHA functional classification. There were good correlations (*r* > 0.4) between LMR (*r* = −0.667, *p* = 0.000), NLR, WMR, HCR, NT-proBNP (*r* = −0.681, *p* = 0.000) and NYHA functional classification, and LMR (AUC = 0.803, 95% CI 0.729–0.849, *p* = 0.0001), NLR and NT-proBNP (AUC = 0.805, 95% CI 0.738–0.861, *p* = 0.0001) had good diagnostic values (AUC > 0.7) for HF in patients with HFpEF. In addition, there were certain correlations between LMR, NT-proBNP and echocardiography indicators of cardiac structural. *Conclusions:* SII have a potential application value in the clinical evaluation of patients with HFpEF in the follow-up, especially in areas with limited medical resources, as they are more convenient and cost effective. Among different SII, LMR is probably the most promising metric. However, large-scale clinical trials are needed in the future to confirm these findings.

## 1. Introduction

Heart failure (HF) is a complex clinical syndrome that results primarily from a variety of serious cardiovascular diseases, and the global prevalence is more than 23 million [1]. Due to its high rehospitalization and mortality rates, HF has become one of the major clinical and public health problems worldwide. The prevalence of HF with preserved ejection fraction (HFpEF), which has similar mortality and possibly worse prognosis compared to HF with reduced ejection fraction (HFrEF), increased by 10% every 10 years due to population aging and growing incidence rates of the diseases related to HFpEF [2]. Furthermore, it is considered that there is no certain correlation between HFpEF and HFrEF, and they are not different courses of a disease [2]. Therefore, the prevention, therapy and management of HFpEF is a challenging work [3].

It is particularly important to make a judgment on patients’ condition through of convenient and economical objective testing, so that the treatment plan can be adjusted in time, as the current treatment of HF is mainly drug based and outside the hospital. Left ventricular ejection fraction (LVEF) and N-terminal pro-B-type natriuretic peptide (NT-proBNP), the two important evaluation indexes of cardiac function, have some limitations in the use of assessing cardiac function in the follow-up of patients with HFpEF. For example, EF values for HfpEF do not decrease, NT-proBNP and echocardiography are expensive and require certain medical conditions. Certainly, echocardiography is an important tool in the diagnosis and evaluation of HfpEF and is indispensable in the initial diagnosis of HfpEF, as it is the method of choice to assess diastolic function, which is at the core of the diagnosis of HfpEF. However, patients with HfpEF in areas lacking medical resources need more economical and convenient objective evaluation metrics, especially in long-term follow-up. Although relevant studies have obtained many other indexes for the clinical evaluation of patients, it is often difficult to promote them in clinical practice because of their complexity or high requirements.

The clinical value of systemic inflammatory indexes (SII) such as neutrophil-to-lymphocyte ratio (NLR), platelet-to-lymphocyte ratio (PLR), lymphocyte-to-monocyte ratio (LMR), fibrinogen-to-albumin ratio (FAR), albumin-to-gamma-glutamyl-transferase ratio (AGR), monocyte/high-density-lipoprotein-cholesterol-C (HDL-c) ratio (MHR), white-blood-cell-count-to-mean-platelet-volume ratio (WMR) and high-density-lipoprotein-cholesterol-to-C-reactive protein (CRP) ratio (HCR) in patients with HFpEF were discussed, because they are easy to obtain and have certain clinical value in the diagnosis, therapy and prognosis of cardiovascular diseases [1,4,5,6,7].

## 2. Method

### 2.1. Study Population

Medical charts of inpatients in The First Affiliated Hospital of Jinan University (Guangzhou, China), from July 2018 to March 2022, were retrieved by electronic medical records. Patients with HFpEF and New York Heart Association (NYHA) functional class II–IV were included in the study group, and patients with NYHA functional class I were included in the control group. Inclusion criteria were: (1) age older than 18 years; (2) HFpEF that met the diagnostic criteria of “2017 ACC/AHA/HFSA Guidelines: Management of Heart Failure” [8]; (3) complete physical examination results and medical history. Exclusion criteria were: (1) hospitalization for acute myocardial infarction (AMI); (2) complicated by severe liver and kidney dysfunction, malignant tumors, or chronic, severe diseases of the blood system and immune system; (3) recent used of glucocorticoids or immunosuppressants; (4) acute or chronic infections diseases (such as pneumonia, infective endocarditis or viral myocarditis, etc.) that cause significant changes in the levels of studied biomarkers; (5) patients with other conditions that may lead to changes in study indicators unrelated to the study purpose (such as alcoholics, short-term surgical procedures, other acute diseases, etc.).

### 2.2. Blood Indexes

The cell counting from whole blood, fibrinogen and comprehensive metabolic panel of all included patients were tested by the Clinical Medical Laboratory Center of The First Affiliated Hospital of Jinan University after admission, and the difference between the time of blood test and the time of the diagnosis of NYHA functional classification was no more than 24 h.

### 2.3. Echocardiography

The echocardiography was performed by Department of Clinical Medical Ultrasound of The First Affiliated Hospital of Jinan University after the patients were admitted to the hospital. Left atrial systolic diameter (LADs), left ventricular diastolic diameter (LVDd), left ventricular posterior wall thickness at end-diastole (LVPWd), left ventricular ejection fraction (LVEF) and late-to-early diastolic transmitral flow velocity (A/E) were measured by improved Simpson method. The difference between the time of the detection of echocardiography and the time of the diagnosis of NYHA functional classification was no more than 24 h.

### 2.4. Evaluation of NYHA Functional Classification

NYHA functional classification refers to the classification of cardiac function proposed by NYHA, in 1928. In this study, on the basis of the NYHA functional classification in revision diagnosis of the admitting diagnosis, two experienced cardiologists evaluated and modified the NYHA functional classification of patients based on the symptom description of the cases and the results of the six-minute walk test, cardiac catheterization, cardiac MRI, etc. If the views of the two physicians were inconsistent, consult a third physician.

### 2.5. Statistical Analysis

Normality test was performed for all continuous variables, and the continuous variables that met normal distribution were expressed as mean ± standard deviation ($\underset{x}{¯}$ ± SD), and compared by *t*-test if also homoscedastic. Otherwise, these variables were expressed as median (interquartile range (IQR)) and compared by Mann–Whitney U test. One-Way ANOVA (Tamhaini or Bonferroni) was used to compare continuous variables between multiple groups. The categorical variables were expressed as frequencies and proportions, and the difference between groups was tested by χ² test. Spearman’s and Pearson’s correlation coefficients were used to analyze the correlation between each study index and NYHA functional classification or echocardiographic indexes, respectively. Logistic regression was used to analyze the independent predictor for HF or increased NYHA functional classification, and operating characteristic curve (ROC) was used to evaluate the diagnostic effect of each index for HF. *p* < 0.05 (bilateral) was considered statistically significant. SPSS (version, 27.0; IBM, IL, USA) and MedCalc (trial version, 20.1; MedCalc Software Ltd., Ostend, Belgium) were used for all statistical analyses.

## 3. Results

### 3.1. Baseline Characteristics

In total, 189 patients (including 48 patients with NYHA I in the control group, and 141 patients with NYHA II–IV in the study group) were included for analysis according to our criteria, and patients with a previous diagnosis of HFpEF but a current cardiac function classification of NYHA I were also included in the control group. Most of these patients were admitted for coronary heart disease (CHD), HF and hypertension; therefore, patients in the control group also have a high prevalence of CHD (65%) and a history of previous related drug use. However, as we record past medication history and most patients may not start treatment or be given a well-established treatment plan until after this admission; this is the reason why the use of HF key therapies such as diuretics (19%) and aldosterone receptor antagonist (6%) is rather low even in NYAH IV patients. Patients in the study group were older (69.82 ± 13.12 vs. 59.02 ± 10.89), with a higher percentage of women (49% vs. 29%), and more complicated with hypertension (67% vs. 50%), atrial fibrillation (21% vs. 2%) or diabetes (35% vs. 15%). The baseline characteristics of all patients are shown in Table 1.

### 3.2. Comparison of SII Levels

Comparing differences in study index levels among different NYHA functional classifications, the results showed that there were statistically significant differences in the NLR, PLR, LMR, FAR, AGR, MHR and NT-proBNP levels, but not in WMR, CHR and LVEF levels. With the addition of the NYHA functional classification, NLR and NT-proBNP levels significantly increased and LMR levels significantly decreased, while there was only an increasing tendency in PLR, FAR, AGR and MHR levels. There were statistically significant differences in NLR, PLR, LMR and NT-proBNP levels between NYHA I and NYHA II-IV groups, but not in FAR, AGR, MHR, WMR, CHR and LVEF levels. NLR, PLR and NT-proBNP levels were higher, while LMR levels were lower in the NYHA II–IV group compared with the NYHA I group (Table 2).

### 3.3. Regression Analysis

#### 3.3.1. Regression Analysis for HF

The factors in the baseline characteristics with statistical difference (*p* < 0.5) between NYHA I and NYHA II–IV groups were included in univariate logistic regression analysis, and the results showed that age, hypertension, diabetes, scr, NLR, PLR, LMR, MHR and NT-proBNP are independent predictors for HFpEF. The factors with statistically significant differences (*p* < 0.05) in the univariate logistic regression analysis were subjected to collinear analysis, and the results showed that the variance inflation factors of the equal factors were all <10 and the tolerance was >0.1; therefore, multivariate logistic regression analysis was performed. Discovered by the Change-in-Estimate method, the confounding factors affecting NLR levels were age, diabetes and scr, affecting PLR levels were age and scr, and affecting NT-proBNP levels was age, while no confounding factors were affecting LMR levels. Statistical results showed that NLR (odds ratio (OR) = 1.388, 95% CI 1.031–1.870, *p* = 0.031), LMR (OR = 0.463, 95% CI 0.348–0.617, *p* = 0.000) and NT-proBNP (OR = 1.002, 95% CI 1.000–1.003, *p* = 0.008) levels were independent predictors for HF after adjusting corresponding confounding factors (Table 3).

#### 3.3.2. Regression Analysis for Increased NYHA Functional Classification

The indexes in the baseline characteristics with statistical difference (*p* < 0.5) between different NYHA functional classifications were included in multivariate logistic regression analysis, NLR (OR = 0.848, 95% CI 0.751–0.957, *p* = 0.007), LMR (OR = 2.630, 95% CI 2.016–3.435, *p* = 0.000), FAR (OR = 0.002, 95% CI 0.000–0.237, *p* = 0.011), AGR (OR = 1.629, 95% CI 1.219–2.175, *p* = 0.001), MHR (OR = 0.090, 95% CI 0.028–0.291, *p* = 0.000) and NT-proBNP (OR = 1.000, 95% CI 1.000–1.000, *p* = 0.003) were independent predictors for increased NYHA functional classification after adjusting for confounding factors for age, drinking, diabetes and scr (Table 4).

### 3.4. Correlation Analysis

#### 3.4.1. Correlation Analysis between SII and NYHA Functional Classification

NLR (*r* = 0.459, *p* = 0.000), PLR (*r* = 0.275, *p* = 0.000), FAR (*r* = 0.376, *p* = 0.000), MHR (*r* = 0.251, *p* = 0.001), CHR (*r* = 0.413, *p* = 0.000) and NT-proBNP (*r* = 0.681, *p* = 0.000) levels were significantly positively correlated, LMR (*r* = −0.667, *p* = 0.000) and AGR (*r* = −0.291, *p* = 0.000) levels were significantly anticorrelated, and WMR and LVEF levels were not correlated with NYHA functional classification by Spearman correlation analysis (Table 5). The most relevant indicators for NYHA functional classification are LMR, which best in SII, and NT-proBNP (Figure 1).

#### 3.4.2. Correlation Analysis between SII and Echocardiography Indexes

Correlation analysis of SII and NT-proBNP with LADs, LVDd, LVPWd and A/E, the results showed that LMR levels were anticorrelated with LADs (*r* = −0.359, *p* = 0.000), LVDd (*r* = −0.213, *p* = 0.003) and LVPWd (*r* = −0.180, *p* = 0.013) levels, FAR levels were positively correlated with LVPWd (*r* = 0.154, *p* = 0.045) levels, AGR was anticorrelated with LADs (*r* = −0.283, *p* = 0.000) levels, and NT-proBNP levels were positively correlated with LADs (*r* = 0.315, *p* = 0.000), LVDd (*r* = 0.279, *p* = 0.000) and LVPWd (*r* = 0.156, *p* = 0.041) levels, all of which were with statistically significant differences. Results that were not statistically different were not listed (Table 6).

### 3.5. Diagnosis of HF

In diagnosis of HF, ROC curve analysis showed statistically significant differences in the area under curve (AUC) of NLR (0.753, 95% CI 0.685–0.813, *p* = 0.0001), PLR (0.648, 95% CI 0.575–0.715, *p* = 0.0017), LMR (0.803, 95% CI 0.729–0.849, *p* = 0.0001), FAR (0.667, 95% CI 0.591–0.737, *p* = 0.0009), CHR (0.654, 95% CI 0.556–0.744, *p* = 0.0085), NT-proBNP (0.805, 95% CI 0.738–0.861, *p* = 0.0001), LMR combined with WMR (0.815, 95% CI 0.752–0.868, *p* = 0.0351), and LMR combined with NT-proBNP (0.841, 95% CI 0.778–0.892, *p* = 0.0001), while AGR, MHR, WMR and LVEF had an AUC of 0.562 (95% CI 0.486–0.636), 0.589 (95% CI 0.514–0.660), 0.516 (95% CI 0.442–0.590) and 0.528 (95% CI 0.455–0.601), respectively, with no significantly statistical differences. In addition, the highest AUC value in SII combined diagnosis of HF comes from LMR combined with WMR (0.815, 95% CI 0.752–0.868), and the highest AUC value in SII combined with NT-proBNP diagnosis of HF is from LMR combined with NT-proBNP (0.841, 95% CI 0.778–0.892) (Table 7), and there were no significant differences in AUC between LMR and NT-proBNP/LMR combined with WMR (Table 8, Figure 2). The best separate diagnostic index for HF in SII is LMR (AUC 0.803), with a critical value of 3.4516, diagnostic sensitivity of 69.50%, and specificity of 79.17%.

## 4. Discussion

The purpose of our study is to investigate the potential clinical value of SII in the diagnosis of HF and ongoing evaluation of condition in patients with HFpEF, which select from SII that have been shown in relevant studies to be clinically significant in the diagnosis, progression, and prognosis of HF-related cardiovascular diseases.

Although inflammation contributes to the pathogenesis and progression of HF across the spectrum of HF, a stronger association of inflammatory markers may exist only in the context of HFpEF, which was demonstrated by COACH and BIOSTAT-CHF trials [9,10]. This may be explained in part by the greater burden of comorbidities in patients with HFpEF, such as diabetes, hypertension, chronic obstructive pulmonary disease, obesity, and chronic kidney disease [11]. Thus, the current consensus is that inflammation plays a varying role in all forms of HF. This is the first step to providing a possibly more convenient and economical index for the auxiliary diagnosis and monitoring of HFpEF, especially in areas with limited medical resources, as the cost of a single SII test is usually very low compared to BNP/NT-proBNP. In Guangzhou, China, for example, the price standards published by the government show that the price of a single test for blood cells in each public hospital is around USD 0.2, fibrinogen, albumin, and HDL-c are around USD 0.7–3, CRP is the most expensive at around USD 4–5, while BNP/NT-proBNP is around USD 26–33, and using a rapid detection method adds another USD 11 [12]. The most expensive tests in SII are HCR and FAR, which are only about one-ninth the price of BNP, and most other SII are about one-sixtieth the price of BNP/NT-proBNP. Typically, the waiting time for most SII metrics is a quarter or less of BNP/NT-proBNP. More importantly, SII can usually be derived from the necessary routine test items, meaning that no additional specific tests are needed.

The results of our study showed that LMR, NLR and NT-proBNP were independent predictors for the presence of HF, and LMR, NLR, AGR and NT-proBNP were independent predictors for increased NYHA functional classification in patients with HFpEF. There were good correlations (*r* > 0.4) between LMR, NLR, WMR, CHR, NT-proBNP and NYHA functional classification, and LMR, NLR, NT-proBNP, LMR combined with WMR, and LMR combined with NT-proBNP had good diagnostic values (AUC > 0.7) for HF in patients with HFpEF. In the joint diagnosis of the two indexes, LMR combined with WMR was the most efficient combination in the combined diagnosis of inflammatory indexes, and LMR combined with NT-proBNP was the most efficient combination in the combined diagnosis of SII and NT-proBNP. In addition, there were certain correlations between LMR/NT-proBNP and LADs/LVDd/LVPWd. From the above results, it can be seen that LMR, similar to NT-proBNP, has a strong correlation with NYHA functional classification, certain correlations with LADs, LVDd and LVPWd, and good diagnostic value for determining the presence of HF status in patients with HFpEF. In the patients with HFpEF, the critical value of LMR used to help diagnose HF is 3.4516, below which the patient’s cardiac function classification may be in NYHA II-IV, and a lower LMR value means that the patient has poorer cardiac function classification. In addition, LMR is higher than NT-proBNP in sensitivity (69.50% vs. 67.94%) and lower than NT-proBNP in specificity (79.17% vs. 90.48%). Further research is still needed to elucidate some specific characteristics of SII with relevance to HF.

At present, there are a large number of people with HF in the world, which has brought a huge economic burden to the world’s healthcare systems, and thus, the diagnosis, evaluation, therapy and prognosis of HF are urgent medical problems to be solved. Hence, it is necessary to provide more economical and convenient medical services for patients with HF in areas with poor medical and health conditions. In the clinical application of condition assessment of patients with HF, the subjective feeling of patients and the clinical experience of doctors will have a certain impact on the accuracy of the NYHA functional classification, and the NYHA functional classification is relatively rough and cannot reflect the continuous changes in cardiac function. Although NT-proBNP is a widely used index which can reflect the cardiac function and has a good diagnostic value for HF [3], it requires special tests and higher medical expenses and conditions, the same is true for echocardiography. Despite the determination of inflammatory markers still being restricted to research purposes [13], it makes sense to explore this further. As a cheap, fast and widely applicable test item, SII can be performed at all levels of medical units, and multi-disciplinary utilization of its value can better serve patients with HF.

HF is not just a problem of the heart itself, but a complex systemic disease from a pathophysiological point of view. Studies confirm that an increasing number of factors are associated with the presence and development of HF, among which the activation of the immune system and the production of inflammation are the focus of attention [14,15,16]. Furthermore, inflammation, a major factor in advanced cardiovascular diseases, is well characterized in the adverse progression of HF of different etiologies. High levels of inflammatory mediators and various blood cell infiltrations have been shown in the circulating and cardiac tissue of patients with cardiovascular diseases, especially HF. Various inflammatory cells infiltrating the heart, which have been confirmed to be closely related to cardiovascular diseases [5,17,18,19,20,21], can lead to HF by producing and secreting various cytokines, regulating the inflammatory response, and affecting the function of other cells and the process of myocardial extracellular matrix remodeling [17,22]. Among these inflammatory cells, lymphocytes and monocytes can be activated by pro-inflammatory cytokines [23,24,25], which convert cells into potential sources of pro-inflammatory cytokines and lead to further activation of these cells, finally leading to systemic inflammation in HF. High circulating levels of pro-inflammatory cytokines may have multiple adverse effects on HF, such as myocardial remodeling, increased arrhythmias and negative inotropes [25,26]. For example, monocytes are a major driver of [18] inflammatory and fibrotic processes in heart diseases and HF [18,27,28]. During the inflammatory response to cardiac injury, monocytes are attracted from the peripheral circulation by chemotactic signals secreted by the endothelium and the injury site, and migrate into tissues to form monocyte phagocyte precursors, which subsequently differentiate into macrophages with different functional properties [29]. Increased monocyte activation is seen in the early and late stages of cardiac disease, and monocyte/macrophage infiltration is often involved in inflammation, fibrosis, endothelial damage, oxidative stress in the myocardium, myocardial fibrosis in the heart, diastolic dysfunction, and tissue damage and repair in HF [30,31,32].

Increased numbers of lymphocytes and macrophages (derived from monocytes) were found in blood and myocardium tissue in a mouse model and humans with HFpEF [33,34], leading to cardiac hypertrophy and fibrosis [35]. A lower relative lymphocyte count and an increased monocyte count are associated with poorer prognosis in HF [17,36,37,38,39,40]. Currently, relative lymphopenia is thought to reflect a response to physiological stress, and mononucleosis reflects chronic systemic inflammation, both of which play prominent roles in inflammation, cardiac remodeling and fibrosis of HF. A lower value of LMR is associated with poor prognosis in atherosclerotic disease [19] and a higher risk of mortality within 6 months of discharge from acute HF [17].

In addition, some cardiac comorbidities relevant to this study need to be explored, as the systemic inflammatory state caused by these comorbidities has recently been shown to be predictive of HFpEF [11]. The kidney and heart are interdependent, and cardiac and renal dysfunction may worsen each other through multiple mechanisms, so that worse cardiac function may be accompanied by diminished renal function, as well as in HFpEF [41]. In our study, considering that deteriorating renal function leads to a variety of hematologic disturbances that severely affect study metrics [42], we excluded patients diagnosed with clinical renal failure; however, higher serum creatinine was still found in NYAH IV patients. The microvascular inflammation hypothesis (MIH) postulates that pro-inflammatory comorbidities lead to low-grade systemic and coronary microvascular endothelial inflammation, myocardial inflammation and subsequent microvascular dysfunction and cardiac fibrosis [11,43]. Comorbidity-driven microvascular inflammation is thought to be the unifying pathophysiologic mechanism in HFpEF. Obesity is a common and strong risk factor in HFpEF because it is pro-inflammatory, and inflammation mediates obesity-associated metabolic disturbances, and elevation of some inflammatory biomarkers is more common in the obese population [44]. Since there was no statistical difference in BMI between the groups of patients in this study, which may be due to the insufficient number of patients included, no further analysis was performed. Diabetes is associated with a 2- to 4-fold increased risk for HF; diabetes has the ability to induce a systemic inflammatory state; and chronic hyperglycemia, hyperinsulinemia and insulin resistance can lead to altered vascular homeostasis, decreased nitric oxide and increased active oxygen levels, which can activate pro-inflammatory pathways that lead to vascular damage and myocardial dysfunction [45]. Previous studies have also shown that some SII are altered in and associated with the onset and development of diabetes; for example, NLR is positively associated with the incidence of diabetes secondary to exocrine pancreatic disease and insulin resistance [46]. The prevalence of diabetes in the study group was significantly higher than that in the control group, so we analyzed diabetes as a confounding factor to reduce the impact on the results.

## 5. Limitations

The limitations of this study are that: firstly, it cannot reveal the exact pathophysiological mechanism behind the findings of this study; secondly, our diagnosis was based on a clinical record system, which means that our included studies were influenced by the clinician’s experience, although the diagnosis was revised; and finally, this study was retrospective from a single center, and the sample size was small. For this reason, external validation of the results obtained in this study in HFpEF by an independent cohort is necessary. Nonetheless, our primary goal was to demonstrate that SII are a readily available, widely used, and inexpensive tool for the auxiliary diagnosis of HF and evaluation of the condition in patients with HFpEF, especially LMR.

## 6. Conclusions

SII have a potential clinical application value in the clinical evaluation of patients with HFpEF in the follow-up, especially in areas with limited medical resources, as they are more convenient and cost effective. Among different SII, LMR is probably the most promising metric. However, large-scale clinical trials are needed in the future to confirm these findings.

## Figures and Tables

**Figure 1 medicina-58-01473-f001:**
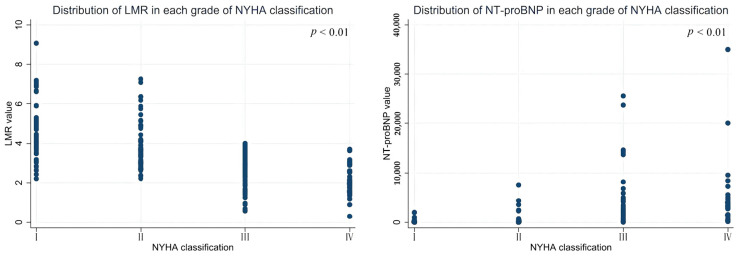
Distribution of LMR and NT-proBNP in different NYHA functional classifications.

**Figure 2 medicina-58-01473-f002:**
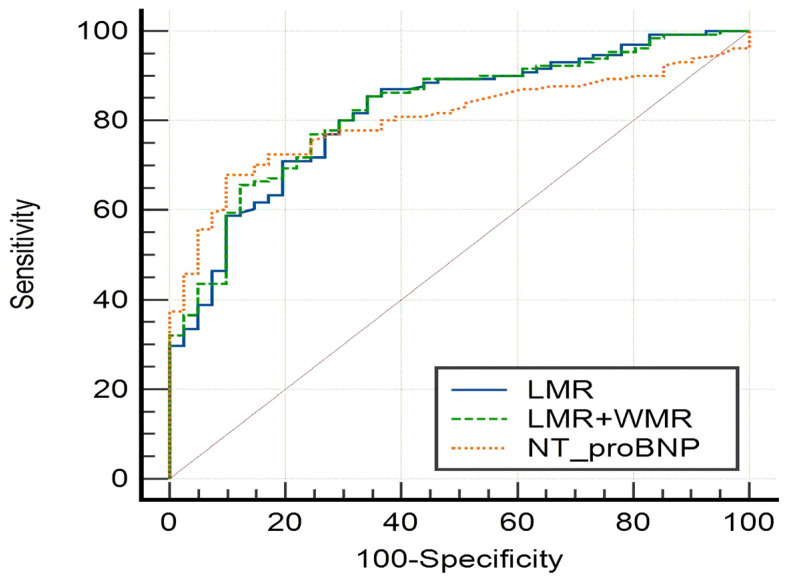
ROC curve analysis of LMR, NT-proBNP and LMR combined with WMR.

**Table 1 medicina-58-01473-t001:** Baseline characteristics of all patients.

Characteristics	NYHA I (*n* = 48)	NYHA II (*n* = 53)	NYHA III (*n* = 56)	NYHA IV (*n* = 32)	*p*	NYHA II–IV (*n* = 141)	*p*
Age (years)	59.02 ± 10.89	66.87 ± 11.38	72.64 ± 13.57	69.77 ± 14.37	0.000 *	69.82 ± 13.12	0.000 *
Male, *n* (%)	34 (71)	28 (53)	27 (48)	17 (53)	0.097	72 (51)	0.017 *
BMI (kg/m^2^)	24.90 ± 3.21	25.02 ± 3.40	24.23 ± 5.00	24.42 ± 10.05	0.831	24.59 ± 5.85	0.239
Smoker, *n* (%)	6 (13)	9 (17)	6 (11)	4 (13)	0.802	19 (13)	0.863
heart rate, (*n*/min)	74.52 ± 10.87	79.38 ± 16.45	75.29 ± 13.00	87.09 ± 19.13	0.001 *	79.51 ± 16.38	0.051
Alcohol drinker, *n* (%)	2 (4)	7 (13)	0 (0)	3 (9)	0.031 *	10 (7)	0.473
WBC (×10^9^/L)	6.73 ± 1.53	6.91 ± 2.04	6.82 ± 2.22	7.81 ± 3.09	0.344	7.08 ± 2.40	0.809
LYM (×10^9^/L)	1.97 ± 0.68	1.74 ± 0.62	1.30 ± 0.49	1.37 ± 0.49	0.000 *	1.48 ± 0.58	0.000 *
MONO (×10^9^/L)	0.46 ± 0.15	0.46 ± 0.16	0.54 ± 0.20	0.88 ± 1.33	0.029 *	0.59 ± 0.67	0.182
NEUT (×10^9^/L)	4.07 ± 1.25	4.53 ± 1.78	4.76 ± 2.03	5.49 ± 2.68	0.019 *	4.84 ± 2.13	0.066
PLT (×10^9^/L)	228.28 ± 52.40	222.69 ± 54.53	200.82 ± 58.36	224.19 ± 110.98	0.077	214.29 ± 72.85	0.222
CRP (mg/dL)	5.17 ± 11.78	4.89 ± 8.28	7.49 ± 13.31	33.59 ± 45.86	0.076	12.51 ± 26.33	0.158
PLT volume (fL)	9.22 ± 0.84	8.99 ± 1.17	9.41 ± 1.45	9.37 ± 1.36	0.307	9.24 ± 1.33	0.740
ALB (g/L)	40.58 ± 6.98	40.85 ± 3.75	36.57 ± 4.62	36.29 ± 4.48	0.000 *	38.17 ± 4.75	0.010 *
γ-GGT (U/L)	34.00 ± 38.53	28.66 ± 19.68	34.50 ± 26.03	78.59 ± 75.17	0.007 *	41.63 ± 44.35	0.308
HDL-c (mmol/L)	1.16 ± 0.39	1.15 ± 0.28	1.07 ± 0.35	1.13 ± 0.29	0.384	1.11 ± 0.31	0.429
FIB (g/L)	3.22 ± 0.81	3.35 ± 0.62	3.55 ± 1.16	4.18 ± 1.33	0.007 *	3.61 ± 1.07	0.035 *
Scr (umol/L)	79.42 ± 23.85	79.30 ± 26.41	112.64 ± 61.76	112.24 ± 49.02	0.000 *	99.84 ± 50.29	0.015 *
AST (U/L)	23.44 ± 5.87	23.62 ± 9.82	24.82 ± 11.73	46.45 ± 51.97	0.103	29.19 ± 27.64	0.952
CKMB (U/L)	20.39 ± 37.39	18.67 ± 6.89	16.96 ± 10.57	23.94 ± 19.55	0.523	19.21 ± 12.37	0.750
CTNI (ng/L)	0.13 ± 0.74	0.01 ± 0.01	0.01 ± 0.01	0.37 ± 0.91	0.317	0.06 ± 0.35	0.482
NT-proBNP	159.10 ± 321.23	572.56 ± 1371.97	3376.93 ± 5440.12	4880.40 ± 6863.44	0.000 *	2715.09 ± 5115.04	0.000 *
LAD (mm)	35.31 ± 4.86	37.36 ± 7.47	44.77 ± 10.99	46.50 ± 10.77	0.000 *	42.38 ± 10.47	0.000 *
LVDd (mm)	44.15 ± 3.80	43.92 ± 4.97	46.64 ± 6.47	48.31 ± 9.22	0.010 *	46.00 ± 6.89	0.254
LVPWd (mm)	8.88 ± 1.53	9.15 ± 1.57	9.59 ± 1.78	9.97 ± 2.24	0.028 *	9.51 ± 1.84	0.032 *
LVEF	62.00 ± 2.79	62.34 ± 4.44	61.55 ± 5.67	62.25 ± 8.20	0.921	62.01 ± 7.79	0.553
A/E (>1)	36 (75)	39 (74)	42 (75)	28 (88)	0.467	109 (77)	0.744
CHD, *n* (%)	31 (65)	41 (77)	37 (66)	18 (56)	0.221	96 (68)	0.655
Hypertension, *n* (%)	24 (50)	36 (68)	36 (64)	22 (69)	0.215	94 (67)	0.039 *
Diabetes, *n* (%)	6 (13)	20 (38)	22 (39)	8 (25)	0.010 *	50 (35)	0.003 *
Hyperlipidemia, *n* (%)	4 (8)	9 (17)	6 (11)	1 (3)	0.218	16 (11)	0.558
Af, *n* (%)	1 (2)	4 (8)	15 (27)	11 (36)	0.000 *	30 (21)	0.002 *
Af with A/E (>1)	1 (2)	3 (6)	13 (23)	10 (31)	0.000 *	26 (18)	0.005 *
ACEI or ARB, *n* (%)	18 (38)	17 (32)	13 (23)	10 (31)	0.463	40 (28)	0.236
Beta-blocker, *n* (%)	16 (33)	12 (23)	14 (42)	10 (22)	0.605	36 (26)	0.296
Diuretics, *n* (%)	1 (2)	1 (2)	3 (5)	6 (19)	0.006 *	10 (7)	0.200
Aldosterone Antagonist, *n* (%)	0 (0)	0 (0)	2 (4)	2 (6)	0.145	4 (3)	0.238
Statin, *n* (%)	6 (13)	8 (15)	7 (13)	7 (22)	0.635	22 (16)	0.601
Metformin, *n* (%)	6 (13)	13 (25)	6 (11)	3 (9)	0.129	22 (16)	0.601

*, *p* < 0.05; BMI, body mass index; WBC, white blood cell; LYM, lymphocyte; MONO, monocyte; NEUT, neutrophil; PLT, platelet; CRP, C reactive protein; ALB, albumin; γ-GGT, gamma-glutamyl transferase; HDL-c, High density lipoprotein cholesterol c; FIB, fibrinogen; NT-proBNP, N-terminal pro-B-type natriuretic peptide; Scr, serum creatinine; AST, aspartate aminotransferase; CKMB, creatine phosphokinase isoenzyme; CTNI, troponin I; LADs, left atrial systolic diameter; LVDd, left ventricular end-diastolic diameter; LVPWd, left ventricular posterior wall thickness at end-diastole; A/E, late and early diastolic mitral valve flow ratio; LVEF, left ventricular ejection fraction; CHD, coronary heart disease; Af, atrial fibrillation; ACEI, angiotensin-converting enzyme inhibitor; ARB, angiotensin II receptor blocker.

**Table 2 medicina-58-01473-t002:** Comparison of SII levels in different cardiac function groups.

Index	NYHA I	NYHA II	NYHA III	NYHA IV	*p*	NYHA II–IV	*p*
NLR NLR	2.389 ± 1.532	3.086 ± 2.937	4.235 ± 2.856	4.341 ± 2.344	0.000 *	3.827 ± 2.821	0.000 *
PLR	131.370 ± 55.964	144.834 ± 71.566	175.866 ± 81.772	169.395 ± 69.441	0.007 *	162.861 ± 76.157	0.010 *
LMR	4.484 ± 1.447	3.946 ± 1.247	2.592 ± 0.908	2.087 ± 0.768	0.000 *	2.986 ± 1.276	0.000 *
FAR	0.096 ± 0.115	0.083 ± 0.188	0.097 ± 0.034	0.118 ± 0.046	0.001 *	0.096 ± 0.035	0.967
AGR	1.769 ± 0.920	2.083 ± 1.224	1.609 ± 0.916	0.934 ± 0.772	0.000 *	1.650 ± 1.104	0.521
MHR	0.444 ± 0.217	0.436 ± 0.219	0.561 ± 0.289	0.819 ± 1.129	0.021 *	0.5711 ± 0.591	0.151
WMR	0.731 ± 0.182	0.791 ± 0.290	0.741 ± 0.270	0.871 ± 0.423	0.257	0.789 ± 0.319	0.739
HCR	5.012 ± 11.924	4.779 ± 8.788	6.298 ± 8.628	30.145 ± 41.964	0.105	11.219 ± 23.563	0.186
LVEF	62.000 ± 2.791	62.340 ± 4.437	61.545 ± 5.666	62.250 ± 8.195	0.921	62.007 ± 7.791	0.553
NT-proBNP	159.119 ± 321.226	572.555 ± 1371.969	3376.926 ± 5440.117	4880.400 ± 6863.435	0.000 *	2715.085 ± 5115.039	0.000 *

*, *p* < 0.05; NLR, neutrophil-to-lymphocyte ratio; PLR, platelet-to-lymphocyte ratio; LMR, lymphocyte-to-monocyte ratio; FAR, fibrinogen-to-albumin ratio; AGR, albumin-to-gamma-glutamyl-transferase ratio; MHR, monocyte/high-density-lipoprotein-cholesterol ratio; WMR, white-blood-cell-count-to-mean-platelet-volume ratio; HCR, high-density lipoprotein cholesterol/C-reactive protein; LVEF, left ventricular ejection fraction; NT-proBNP, N-terminal pro-B-type natriuretic peptide.

**Table 3 medicina-58-01473-t003:** Logistic regression analysis for HF.

Index	Univariate			Multivariate		
	OR	95% CI	*p*	OR	95% CI	*p*
Age	1.917	1.038–1.100	0.000 *			
Male	1.917	1.150–4.708	0.019 *			
Hypertension	1.917	0.980–3.748	0.057			
Diabetes	3.846	1.529–9.674	0.004 *			
Scr	1.015	1.004–1.027	0.010 *			
NLR	1.703	1.248–2.325	0.001 *	1.388	1.031–1.870	0.031 *
PLR	1.008	1.002–1.014	0.012 *	1.005	0.998–1.012	0.138
LMR	0.463	0.348–0.617	0.000 *	0.463	0.348–0.617	0.000 *
NT-proBNP	1.002	1.001–1.004	0.004 *	1.002	1.000–1.003	0.008 *

*, *p* < 0.05; CI, confidence interval; NLR, neutrophil-to-lymphocyte ratio; PLR, platelet-to-lymphocyte ratio; LMR, lymphocyte-to-monocyte ratio; NT-proBNP, N-terminal pro-B-type natriuretic peptide.

**Table 4 medicina-58-01473-t004:** Multivariate regression analysis of predictors for increased NYHA functional classification.

Index	OR	95% CI	*p*
NLR	0.848	0.751–0.957	0.007 *
PLR	0.996	0.993–1.000	0.079
LMR	2.630	2.016–3.435	0.000 *
FAR	0.002	0.000–0.237	0.011 *
AGR	1.629	1.219–2.175	0.001 *
MHR	0.090	0.028–0.291	0.000 *
NT-proBNP	1.000	1.000–1.000	0.003 *

*, *p* < 0.05; CI, confidence interval; NLR, neutrophil-to-lymphocyte ratio; PLR, platelet-to-lymphocyte ratio; LMR, lymphocyte-to-monocyte ratio; FAR, fibrinogen-to-albumin ratio; AGR, albumin-to-gamma-glutamyl-transferase ratio; MHR, monocyte/high-density-lipoprotein-cholesterol ratio; NT-proBNP, N-terminal pro-B-type natriuretic peptide.

**Table 5 medicina-58-01473-t005:** Spearman correlation analysis of SII and NYHA functional classification.

Spearman	NLR	PLR	LMR	FAR	AGR	MHR	WMR	CHR	LVEF	NT-proBNP
*r*	0.459	0.275	−0.667	0.376	−0.291	0.251	0.016	0.413	−0.104	0.681
*p*	0.000 *	0.000 *	0.000 *	0.000 *	0.000 *	0.001 *	0.832	0.000 *	0.156	0.000 *

*, *p* < 0.05; NLR, neutrophil-to-lymphocyte ratio; PLR, platelet-to-lymphocyte ratio; LMR, lymphocyte-to-monocyte ratio; FAR, fibrinogen-to-albumin ratio; AGR, albumin-to-gamma-glutamyl-transferase ratio; MHR, monocyte/high-density-lipoprotein-cholesterol ratio; WMR, white-blood-cell-count-to-mean-platelet-volume ratio; HCR, high-density lipoprotein cholesterol/C-reactive protein; LVEF, left ventricular ejection fraction; NT-proBNP, N-terminal pro-B-type natriuretic peptide.

**Table 6 medicina-58-01473-t006:** Pearson correlation analysis of SII and echocardiography indexes.

Index	LADs	LVDd	LVPWd	Pearson
LMR	−0.359	−0.213	−0.180	*r*
	0.000 *	0.003 *	0.013 *	*p*
FAR			0.154	*r*
			0.045 *	*p*
AGR	−0.283			*r*
	0.000 *			*p*
NT-proBNP	0.315	0.279	0.156	*r*
	0.000 *	0.000 *	0.041 *	*p*

*, *p* < 0.05; LMR, lymphocyte-to-monocyte ratio; FAR, fibrinogen-to-albumin ratio; AGR, albumin-to-gamma-glutamyl-transferase ratio; NT-proBNP, N-terminal pro-B-type natriuretic peptide; LADs, left atrial systolic diameter; LVDd, left ventricular end-diastolic diameter; LVPWd, left ventricular posterior wall thickness at end-diastole; LVEF, left ventricular ejection fraction; A/E, late-to-early diastolic transmitral flow velocity.

**Table 7 medicina-58-01473-t007:** ROC curve analysis of SII for diagnosis of HFpEF.

Index	Youden	Critical Value	Sensitivity	Specificity	AUC	*p*	95% CI
NLR	0.4681	2.1563	78.72	68.09	0.753	0.0001 *	0.685–0.813
PLR	0.3054	123.1707	65.96	64.58	0.648	0.0017 *	0.575–0.715
LMR	0.4867	3.4516	69.50	79.17	0.803	0.0001 *	0.729–0.849
FAR	0.2842	0.0807	57.69	70.73	0.667	0.0009 *	0.591–0.737
AGR	0.1985	0.7521	26.67	93.18	0.562	0.1762	0.486–0.636
MHR	0.2184	0.2529	89.93	31.91	0.589	0.0737	0.514–0.660
WMR	0.1986	0.9519	30.50	89.36	0.516	0.7093	0.442–0.590
CHR	0.3581	1.5248	67.95	67.86	0.654	0.0085 *	0.556–0.744
LVEF	0.1999	59	26.24	93.75	0.528	0.5002	0.455–0.601
NT-proBNP	0.5842	219	67.94	90.48	0.805	0.0001 *	0.738–0.861
LMR+WMR					0.815	0.0351 *	0.752–0.868
LMR+NT-proBNP					0.841	0.0001 *	0.778–0.892

*, *p* < 0.05; CI, confidence interval; AUC, area under curve; NLR, neutrophil-to-lymphocyte ratio; PLR, platelet-to-lymphocyte ratio; LMR, lymphocyte-to-monocyte ratio; FAR, fibrinogen-to-albumin ratio; AGR, albumin-to-gamma-glutamyl-transferase ratio; MHR, monocyte/high-density-lipoprotein-cholesterol ratio; WMR, white-blood-cell-count-to-mean-platelet-volume ratio; HCR, high-density lipoprotein cholesterol/C-reactive protein; LVEF, left ventricular ejection fraction; NT-proBNP, N-terminal pro-B-type natriuretic peptide.

**Table 8 medicina-58-01473-t008:** Differences in AUC between LMR and NT-proBNP or LMR combined with WMR.

Index	AUC	Difference	SE	95% CI	z	*p*
LMR/NT-proBNP	0.811/0.805	0.00618	0.0430	−0.0781–0.0905	0.144	0.8858
LMR/LMR+WMR	0.803/0.815	0.00407	0.0054	−0.00651–0.0147	0.755	0.4505

CI, confidence interval; AUC, area under curve; SE, standard error; LMR, lymphocyte-to-monocyte ratio; WMR, white-blood-cell-count-to-mean-platelet-volume ratio; NT-proBNP, N-terminal pro-B-type natriuretic peptide.

## Data Availability

Data will be made available on reasonable request.

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
