# Peer review of "Application Value of Systemic Inflammatory Indexes in the Clinical Evaluation of Patients with Heart Failure with Preserved Ejection Fraction (HFpEF)"

_medicina, 2022, doi:10.3390/medicina58101473_

Round 1

Reviewer 1 Report

MS: Application value of systemic inflammatory indexes in the diagnosis and evaluation of the cardiac function in patients with HFpEF

This MS is interesting since it presents the clinical application of systemic inflammatory indexes (SII) in diagnosing and evaluating cardiac functions in patients with HFpEF. This is particularly important in areas with limited medical resources. This MS emphasizes the value of lymphocyte to monocyte ratio (LMR) as the more convenient and cost-effective potential alternative to NT-proBNP.

The Authors can consider some suggestions for modifications/corrections of the text:

P # 1, MS title

Application value of systemic inflammatory indexes in the diagnosis and evaluation of the cardiac function in patients with heart failure with preserved ejection fraction (HFpEF)

The Authors may consider a modification of the MS title

P # 1 Abstract

Conclusions:

SII have a potential clinical application value in the diagnosis and evaluation of cardiac function in patients with HFpEF, especially in areas with limited medical resources. Among different SII, LMR is the more convenient and cost-effective alternative to NT-proBNP. However, large-scale clinical trials are needed in the future to confirm these findings.

The Authors may consider a modification of the Conclusions.

P # 1 INTRODUCTION

The prevalence of HF with preserved ejection fraction (HFpEF), which has similar mortality and possibly worse prognosis, compared to HF with reduced ejection fraction (HFrEF), increased by 10% every 10 years due to the population aging and growing incidence rates of the diseases related to HFpEF.

The Authors may consider a modification of the INTRODUCTION.

Multiple, similar modifications for clarity, grammar, and style need to be done in the whole text.

P # 2 METHOD

2.1. The Study Group

Medical charts of inpatients in The First Affiliated Hospital of Jinan University [please, specify a city, country] from July 2018 to March 2022 were retrieved by electronic medical records. Patients diagnosed with HFpEF were included in the study (experimental) group and patients with New York Heart Association (NYHA) functional class â…  were included in the control group. The diagnostic criteria for HFpEF in this study refer to the “2017 ACC/AHA/HFSA Guidelines: Management of Heart Failure” (Yancy 79 et al., 2017). Inclusion criteria were: (1) age older than 18 years; (2) HFpEF that meets the diagnostic criteria of relevant guidelines; (3) complete physical examination results and medical history. Exclusion criteria were: (1) hospitalization for an acute myocardial infarction (AMI); (2) HF complicated by severe liver and kidney dysfunctions, malignant tumors, or chronic, severe diseases of the blood system and immune system; (3) recent use of glucocorticoids or immunosuppressants; (4) acute or chronic infections (such as lung infections) that cause significant changes in the levels of studied biomarkers.

The Authors may consider a modification of the METHOD section

P # 11, 12 DISCUSSION

At present, there are a large number of people with HF in the world, which has brought a huge economic burden to the world healthcare systems. Hence the diagnosis, evaluation, therapy, prognosis, and outcome of HF represent urgent medical problems to be solved.

The purpose of our study is to explore whether SII are potentially useful clinical parameters for the diagnosis and ongoing evaluation of cardiac functions in patients with HFpEF. This is the first step to providing a possibly more convenient and economical index for the diagnosis and monitoring of HFpEF, especially in areas with limited medical resources. Further research is still needed to elucidate some specific characteristics of SII with relevance to HF.

The Authors may consider a modification of the DISCUSSION section

P # 13 Conclusions

However, future clinical trials on large patient populations are needed to confirm these results.

The Authors may consider a modification of the Conclusions.

Thank you.

Author Response

First of all, I am very grateful to the reviewer for your efforts. The following are my answers to the questions raised by the reviewer.

MS: Application value of systemic inflammatory indexes in the diagnosis and evaluation of the cardiac function in patients with HFpEF

This MS is interesting since it presents the clinical application of systemic inflammatory indexes (SII) in diagnosing and evaluating cardiac functions in patients with HFpEF. This is particularly important in areas with limited medical resources. This MS emphasizes the value of lymphocyte to monocyte ratio (LMR) as the more convenient and cost-effective potential alternative to NT-proBNP.

The Authors can consider some suggestions for modifications/corrections of the text:

P # 1, MS title

Application value of systemic inflammatory indexes in the diagnosis and evaluation of the cardiac function in patients with heart failure with preserved ejection fraction (HFpEF)

The Authors may consider a modification of the MS title

It has been modified according to your suggestion, line 2-4.

P # 1 Abstract

Conclusions:

SII have a potential clinical application value in the diagnosis and evaluation of cardiac function in patients with HFpEF, especially in areas with limited medical resources. Among different SII, LMR is the more convenient and cost-effective alternative to NT-proBNP. However, large-scale clinical trials are needed in the future to confirm these findings.

The Authors may consider a modification of the Conclusions.

It has been modified according to your suggestion, line 39-44, 166-170.

P # 1 INTRODUCTION

The prevalence of HF with preserved ejection fraction (HFpEF), which has similar mortality and possibly worse prognosis, compared to HF with reduced ejection fraction (HFrEF), increased by 10% every 10 years due to the population aging and growing incidence rates of the diseases related to HFpEF.

The Authors may consider a modification of the INTRODUCTION.

Multiple, similar modifications for clarity, grammar, and style need to be done in the whole text.

It has been modified according to your suggestion, line 52-56.

Revisions have been made to the full text.

P # 2 METHOD

2.1. The Study Group

Medical charts of inpatients in The First Affiliated Hospital of Jinan University [please, specify a city, country] from July 2018 to March 2022 were retrieved by electronic medical records. Patients diagnosed with HFpEF were included in the study (experimental) group and patients with New York Heart Association (NYHA) functional class â…  were included in the control group. The diagnostic criteria for HFpEF in this study refer to the “2017 ACC/AHA/HFSA Guidelines: Management of Heart Failure” (Yancy 79 et al., 2017). Inclusion criteria were: (1) age older than 18 years; (2) HFpEF that meets the diagnostic criteria of relevant guidelines; (3) complete physical examination results and medical history. Exclusion criteria were: (1) hospitalization for an acute myocardial infarction (AMI); (2) HF complicated by severe liver and kidney dysfunctions, malignant tumors, or chronic, severe diseases of the blood system and immune system; (3) recent use of glucocorticoids or immunosuppressants; (4) acute or chronic infections (such as lung infections) that cause significant changes in the levels of studied biomarkers.

The Authors may consider a modification of the METHOD section

It has been modified according to your suggestion, line 85-99.

P # 11, 12 DISCUSSION

At present, there are a large number of people with HF in the world, which has brought a huge economic burden to the world healthcare systems. Hence the diagnosis, evaluation, therapy, prognosis, and outcome of HF represent urgent medical problems to be solved.

The purpose of our study is to explore whether SII are potentially useful clinical parameters for the diagnosis and ongoing evaluation of cardiac functions in patients with HFpEF. This is the first step to providing a possibly more convenient and economical index for the diagnosis and monitoring of HFpEF, especially in areas with limited medical resources. Further research is still needed to elucidate some specific characteristics of SII with relevance to HF.

The Authors may consider a modification of the DISCUSSION section

It has been modified according to your suggestion, line 329-333, 371-378.

P # 13 Conclusions

However, future clinical trials on large patient populations are needed to confirm these results.

The Authors may consider a modification of the Conclusions.

Thank you.

It has been modified according to your suggestion, line 166-170.

These are my answers to the reviewer's questions, and thanks again to the reviewers for such a careful revision.

Reviewer 2 Report

The article deals with the interesting issue of inflammatory markers in HFpEF. The topic is interesting, as the prevalence of HFpEF is increasing globally and its pathophysiology is still under investigation. A strength of the present study is the focus on areas of the world with low economic resources. This is a monocentric retrospective longitudinal study that compares hospitalized patients with HFpEF with NYHA functional class >I and control patients with NYHA I hospitalized for other reasons.

Despite the potential interesting topic, in my opinion the study presents several limitations:

1.     Many inaccuracies exist in the design of the study. In fact, in many parts of the text the Authors state that inflammatory markers are “risk factors” for HFpEF (for example, page 5 line 176), but this represent a methodological error. In fact, to the best pf my knowledge, a retrospective longitudinal study only can identify factors that are ASSOCIATED to the investigated disease (in this case, HFpEF), and not risk factors for developing the disease itself. This is particularly evident because, as stated in the Method section, lab test was obtained almost contemporarily to the diagnosis of HFpEF, so the disease was already present at the time of the determination of the “risk factor”. This is evident for NTproBNP: Authors define it as a risk factor for developing HF, but it is extensively reported in literature that NTproBNP is a marker of HF and not a risk factor for developing HF. Therefore, the fact that international guidelines attribute strong value for HFpEF diagnosis to NTproBNP and that determination of inflammatory markers is still restricted to research purpose should be clearly cited in the text.

2.     Inflammatory markers are extremely non-specific (and especially those used in the study, e.g. LMR), while NT-proBNP has a clear pathophysiological link with HF. The Authors should describe the reasons by which inflammatory markers can be used to diagnose HFpEF in the same way as NTproBNP, as they propose. Without discussing this issue, the value of the study may appear low. In addition, further discussion on why these specific SII (e.g. LMR) and not others have been used in this study seems appropriate to understand the main scope of the study.

3.     Patients’ characteristics need to be described more accurately. Patients hospitalized for reasons other than HF can present other diseases that determine elevation of inflammatory markers, apart from those identified in Methods section, i.e. may have undergone surgical procedures. In addition, alcoholism can induce elevation of PCR and alteration in WBC count, and a relevant quote of the patients is described as alcohol drinkers (26/189). Moreover, in many areas of the world, and especially developing countries, latent infections may determine low level of inflammation, i.e. TB, malaria, etc. In other words, how Authors are certain that the elevation of inflammatory markers attributed to HF and not to other comorbidities or concomitant conditions? This issue also seems to restrict the relevance of this study in clinical practice especially in developing countries where inflammatory markers may be elevated also in general population apart from HFpEF diagnosis.

4.     The role of comorbidities in HF patients is not mentioned. In particular, obesity has been recently identified a drive for inflammation and subsequent HFpEF development and obese-inflammatory phenotypes have been described (reference: Sabbah et al, Circ HF 2020). Apart from that, other studies described a “microvascular inflammation hypothesis” that correlates inflammation and HFpEF (reference: Paulus et al, J Am Cardiology 2013). Similarly, the finding that diabetes is more prevalent in HFpEF patients than control group deserves a more accurate discussion, since diabetes and HFpEF seem to be strictly correlated in pathophysiology.

5.     p. 12 line 348 (“inexpensive”). If Pharmaco-Economics issues are discussed, more data in the results and discussion sections should be given regarding the cost of SII determination compared to BNP/NTproBNP in the specific setting of the study.

6.     The manuscript seems to lack originality with respect to previously published papers, often conducted with bigger sample sizes and more robust design (reference: Albar et al, 2022 March 15, Am J card: “Inflammatory markers and risk of HFpEF”, that particularly investigates the IL6/CRP pathway). Notably, none of previous studies identifies elevation of non-specific inflammatory markers to clear development of HFpEF nor proposes to abandon BNP/NTproBNP measurement in favour to other biomarkers.

7.     The introduction and discussion sections seem not to include reference to seminal works in the field and only cite rather old papers, nor international guidelines on HFpEF, and therefore need to be carefully revised. For example: a) Hannah, 2020, Cardiovascular Drugs and Therapy, “Inflammatory cytokines and chemokines as therapy targets in HF” b) Abernethy, 2018, J Am H Association, “Pro-inflammatory biomarkers in stable and decompensated HF”

8.     The impression is that English language level is quite poor, from the title (“application value”) to the first phrase of the abstract (“make a judgement on cardiac function”: do Authors mean that cardiac function need to be assessed?), to various other sections (in particular 3.2 and 3.4.2 that seem only a mere list of abbreviations and do not add any information to the relative tables)

MAJOR COMMENTS

p. 1 line 35. The conclusion by which LMR is an alternative to NTproBNP for diagnosis of HFpEF reflects a profound inaccuracy in te design of the study. The fact that advanced NYAH class patients present elevated NTproBNP and also an altered LMR does not automatically mean that LMR is a diagnostic tool for HFpEF, especially if no pathophysiological pathways are hypothesised.

p. 2 line 57. Please discuss the issue that echo allows evaluation of diastolic function which is key for diagnosis of HFpEF in contrast to HFrEF

p. 2 line 58. Give reference for the cost of NTproBNP determination. Moreover, specify why Authors focus on NTproBNP instead of BNP

p. 3 line 78. Please define better why NYHA I patients were hospitalized, and please discuss if patients with HFpEF can present in NYAH I. Moreover, discuss why in control group 8% of patients is taking ACE-I and 33% of them is taking beta-blockers

p. 3 line 131. Discuss why the use of HF key therapies such as diuretics and MRA is rather low (<20%) even in NYAH IV patients. In addition, discuss cardiac comorbidities in control group, because NYAH I control group has 65% prevalence of CAD

p. 4 lines 97-99. Diastolic function in HFpEF should be described more in detail than the sole determination of E/A ratio, according to more updated guidelines (i.e. 2O16 EACVI guidelines on diastolic function). Moreover, what about E/A ratio in patients with AF?

p. 4 table 1. Please discuss the finding that serum creatinine is more elevated in NYAH IV patients. HFpEF diagnosis is already well known to be correlated to comorbidities, especially CKD (reference: Van de Wouw, Jens, et al. "Chronic kidney disease as a risk factor for heart failure with preserved ejection fraction: a focus on microcirculatory factors and therapeutic targets." Frontiers in Physiology 2019)

p. 4 section 3.3.31. Please define how regression analysis in a retrospective study can identify correlation and not risk factors

p. 11 DISCUSSION section. Please specify reference values for LMR

p. 11 line 261. The term “occurrence” of HFpEF is not correct. HFpEF already occurred at the time of lab test performance, as stated in the Methods section

p. 11 lines 275-277. Further studies with bigger sample size and more detailed protocols are needed to propose other biochemical alternatives to BNP/NTproBNP. In addition, a retrospective study only can explore correlations and not causative relations between variables. Therefore, this sentence seems profoundly incorrect and should be deleted or accurately revised

p. 11 lines 295. Please cite updated literature, i.e. a) Iqbal 2012 “Cardiac biomarkers as new tools for HF management” b) Gaggin 215 JACC “Biomarkers in HF”. The latter in particular stated that SII are markers of comorbidity and can influence prognosis of HFpEF rather than be a diagnostic tool

MINOR COMMENTS

p. 1 line 11. Cardiac function should be defined by echocardiography, not lab tests

p. 1 line 17. Define “certain”: do Authors mean low? Or rather high?

p. 1 line 18. Better specify if 189 patients are control group or HF patients

p. 1 line 33. Inflammatory markers cannot be defined as a diagnostic methods for diagnosis of HFpEF based on the results in this study; please re-write this sentence

p. 1 line 39. “Serious or late”, why “or”? Please specify this concept or correct

p. 1 line 44. Please give reference for mortality data in HF

p. 2 line 50. Please give reference for the low use of echo in some areas of the world

p. 2 line 67. Give reference for the cost of determination of high-density lipoprotein cholesterol in comparison to BNP

p. 2 line 81. Give reference for “relevant guidelines”

p. 2 line 82. Authors mention myocardial infarction, but what about valve disease?

p. 3 line 111. Please correct “consult a third”

p. 8 fig 1. Please make the figure more clear, adding legends in the figure and p values

p. 8 line 218. Please better specify negative correlation. Low LMR correlates with higher LADs?

p. 9 tab 7. The A/E column is empty and could be erased

p. 11 line 272. Please specify better the concept of diagnostic value

p. 12 line 335. Citing the purpose of the study at the end of the discussion sections seems inaccurate; please move this paragraph at the beginning of the section.

Reviewer 3 Report

No self citation and ethical problems. Marked those selections mistakenly.

Report is an origin al work showing complex inflamatory marker and HFPEF relation and I believe it will be beneficial to readers. MY decision is ACCEPT.

Thanks

Author Response

Thank you very much for the reviewers' approval and for your efforts.

Round 2

Reviewer 2 Report

Despite some major improvements in the overall editing of the manuscript has been performed, some major issues still exist that in my opinion could be improved.

1. Title. Since the diagnosis of HFpEF in the study population has already been made at the moment of SII determination, the title should be corrected. In oher words, in this study the Authors explore markers that are altered in patients that already are diagnosed with HFpEF. Moreover, SII are not used here to determine the cardiac function, but are used to make correlation with NYHA class of HFpEF patients. Therefore, the title could be "Application value of systemic inflammatory indexes in the clinical evaluation of patients with heart failure with preserved ejection fraction (HFpEF)". Reference to "cardiac function" should be erased also in the conclusions section.

2. English grammar and medical terms could be implemented in the whole manuscript (as an example: line 119, “infected valve disease” should be corrected with “infective endocarditis”).

3. Line 175. Please specify why "most patients were admitted for hypertension". Do Authors mean hypertensive emergencies? Orwas hypertension present as a comorbidity?

4. Line 243. Please specify the meaning of the "beta" value that is reported for each statistical correlation, either here or in the Methods section.

5. Line 264. On which basis do Authors define these parameters as "the best correlated index" with cardiac function? Please specify this important concept. Moreover, cardiac function and NYHA class are different entities. Here no reference to cardiac function assessment (i.e. LVEF) is reported.

Author Response

Thank you for the reviewer's comments. Please excuse that I may not be able to correspond the lines marked by the reviewer to some of the questions due to software differences, which may result in incorrect answers.

1. Title.Since the diagnosis of HFpEF in the study population has already been made at the moment of SII determination, the title should be corrected. In oher words, in this study the Authors explore markers that are altered in patients that already are diagnosed with HFpEF. Moreover, SII are not used here to determine the cardiac function, but are used to make correlation with NYHA class of HFpEF patients. Therefore, the title could be "Application value of systemic inflammatory indexes in the clinical evaluation of patients with heart failure with preserved ejection fraction (HFpEF)". Reference to "cardiac function" should be erased also in the conclusions section.

Modify according to your suggestions, e.g. (line 2-4, 36-38), and other places also made corresponding changes.

Amend as follows:

-Application value of systemic inflammatory indexes in the clinical evaluation of patients with heart failure with preserved ejection fraction (HFpEF)

-SII have a potential application value in the clinical evaluation of patients with HFpEF in follow-up,

2. English grammar and medical terms could be implemented in the whole manuscript (as an example: line 119, "infected valve disease"should be corrected with "infective endocarditis").

Modify according to your suggestions (I have tried my best to change the language, and if necessary I will seek the help of a professional company), e.g.

-chang "infected valve disease" to "infective endocarditis"

-chang "infected cardiomyopathy" to "viral myocarditis"

-chang "Lung infection" to "pneumonia"

-chang "Whole blood cell count" to "cell counting from whole blood"

-chang "coagulation function" to "fibrinogen"

-chang "biochemistry" to "comprehensive metabolic panel"

3. Line 175. Please specify why "most patients were admitted for hypertension". Do Authors mean hypertensive emergencies? Orwas hypertension present as a comorbidity?

Hypertension mainly present as co-morbidity, of course, some patients are admitted to the hospital for differential diagnosis of hypertension or for a thorough examination of possible complications, as outpatient visits are not covered by medical insurance.

4. Line 243. Please specify the meaning of the "beta" value that is reported for each statistical correlation, either here or in the Methods section.

This was my mistake and has been changed to OR value and corresponding 95% CI, Table 3, 4.

5. Line 264. On which basis do Authors define these parameters as "the best correlated index" with cardiac function? Please specify this important concept. Moreover, cardiac function and NYHA class are different entities. Here no reference to cardiac function assessment (i.e. LVEF) is reported.

Chang "cardiac function" to "NYHA function classification".

This result is based on the fact that in the Spearman correlation analysis, the absolute value of the correlation coefficient (r) of LMR is the largest in SII, representing the strongest correlation with NYHA function classification.

Above are my answers to the reviewer's questions, thanks again to the reviewer for your efforts.
